

# A Comparison of Contact Charging and Impact Ionization in Low Velocity Impacts: Implications for Dust Detection in Space

Tarjei Antonsen[1], Ingrid Mann[1], Jakub Vaverka[2], Libor Nouzak[2], and Åshild Fredriksen[1]

[1]Department of Physics and Technology, UiT- The Arctic University of Norway, 9037 Tromsø, Norway
[2] Department of Surface and Plasma Science, Faculty of Mathematics and Physics, Charles University, Prague, 180 00, Czech Republic

**Correspondence:** T. Antonsen (tarjei.antonsen@uit.no)

**Abstract.** We investigate the generation of charge during collision of projectiles with sizes below $\sim 1\,\mu$m and metal surfaces at speeds $\sim 0.1$ to $10\,\mathrm{kms}^{-1}$. This corresponds to speeds above the elastic limit and well below speeds where volume ionization can occur. The conditions that we consider apply to dust particles naturally occurring in space and in Earth's upper atmosphere and their direct impacts on rockets, spacecraft, and impacts of secondary ejecta. We introduce a model of capacitive contact

charging in which we allow for projectile fragmentation upon impact, and show that this model describes measurements of metal-metal impacts in the laboratory and in-situ measurements of dust in the Earth's atmosphere well. We have considered the utilization of our model for different scenarios in interplanetary space and in Earth's atmosphere. From this discussion we find it likely that our work can be employed in a number of situations where impact velocities are relatively small. Furthermore, we have discussed the thermodynamics of the low velocity solution of shock wave ionization, and conclude that the impurity

charging effect utilized in the much used model of Drapatz and Michel (1974) does not sufficiently describe charge generation at impact speeds below a few kilometers per second. Consequently, impact charging at low speeds cannot be described with a Saha-solution.

## 1 Introduction

The variables in any experiment studying the impact of dust grains – be it of terrestrial, meteoric, interplanetary or interstellar

origin – span many orders of magnitude. By *variables* we mean the aggregation of ambient parameters and intrisic parameters of the projectile dust grains and impact surfaces. The ambient parameters, such as neutral and charged species densities and temperature can span several orders of magnitude. Combining this fact with the notion that the material properties of the plethora of different dust types that can produce charge in an impact process are also highly variable, there are arguably no single experiments or theoretical considerations that can give a satisfactory explanation of observed phenomena across all

possible combinations of *variables*. In this paper, we focus on what we find is a gap in the knowledge about impact charge production at low impact speeds, i.e. $v_p \lesssim 10\,\mathrm{kms}^{-1}$.

    Early experimental studies of impact ionization of micrometer sized grains on metal surfaces were applied in designing highly sensitive micrometeor detectors; see e.g. Auer and Sitte (1968) and Adams and Smith (1971). The mode of operation of such detectors is to measure the material specific charge generation, which in general yields a semi-empirical relation on the





form: $Q \propto m^{\alpha} v^{\beta}$. This allowed for detection of grains down to sizes $\sim 100$ nm at speeds between the order of 1 kms$^{-1}$ and some tens of kms$^{-1}$. The velocity range is important here, as it has been proposed to be bounded by the limit for production of *ionization* on impact – which we shall understand here as the release of electrons and ions according to a Saha-equation. The pioneering developments in the late 60's and early 70's motivated the employment of impact particle detectors on spacecraft

such as HELIOS, Galileo and Cassini which have all succesfully detected cosmic dust (see e.g. Auer (2012)).

In a treatment of impact charging at speeds up to some tens of kms$^{-1}$, Drapatz and Michel (1974) proposed a mechanism for charge generation by shock wave propagation into both projectile and impact surface. This theory is often referred to as shock wave ionization. This mechanism has been widely used in describing the charge generation in dust accelerators and spacecraft dust impacts. For typical impact speeds in many interplanetary spacecraft and laboratory experiments ($\gtrsim 10$ kms$^{-1}$

), the shock wave model performs very well. However, as discussed in the current work, we find that the proposed theory in its extrapolation down to lower speeds of a few kms$^{-1}$ does not describe the charge generation at low speeds sufficiently. In general, we find that it underestimates the amount of generated charge. There are several applications in the laboratory and in space (satellites, rockets and even spacecraft) that encounter low impact speeds by particles down to nanoscale size where the shock wave ionization theory thus may not be used.

Motivated by the notions above, we propose in this paper a new charging theory for impact speeds below $\sim 10$ kms$^{-1}$ and projectile sizes down to nanoscale. The theory utilizes the concept of contact (capacitive) charging and furthermore includes the parameterization of fragmentation of the projectile particles on impact into a distribution of smaller grains. The resulting charge generation is then dependent on the degree of fragmentation and affinity to exchange charge capacitively; the latter is dependent on the difference in work function between projectile and impact surface. The models of charging and fragmentation

that constitute our novel approach are presented in sections 2.1 and 2.2. The shock wave ionization theory is presented in 2.3. The results from comparisons of charging models in metal-metal and ice-metal collisions are presented in section 3. The thermodynamics and limitations of the shock wave ionization model at low speeds is discussed in 4.1. We furthermore discuss the application areas of a contact charging model for low velocity impacts of dust on metal surfaces.

## 2   Charging and Fragmentation Models

In the sections below, we introduce the theoretical framework for our contact charging model based on fragmentation and capacitive charging, as well as the theory of shock wave ionization with special emphasis on the low velocity regime. We utilize our model on two slightly different types of projectile grains.

The motivation behind model presented below, and approach for its utilization, can be summarized as follows: At speeds comparable to or lower than the ciritical limit for significant deformation or cratering in a grain-surface collision – see e.g.

Jones et al. (1996) – there is little to no available material or energy for impact ionization. Moreover, the incoming projectile grains will fragment, as low energy collisions essentially can be viewed as a collision cascade as opposed to a sublimation-like destruction process. The consequence is that impact ionization models such as shock wave ionization overestimate the produced charge for dust-surface interactions at low velocities. Our solution is to invoke a model in which no direct charge (plasma)





production takes place, but rather takes advantage of the fact that (semi-) conducting grains can have a capacitive coupling to surfaces when there is a difference in effective work function between them. In such a scenario, the charge production is in simple terms due to electrons jumping between the surfaces of fragments and target in an effective potential.

## 2.1 Fragmentation Model

At speeds $\gtrsim 100 \text{ ms}^{-1}$, both metal particles (Froeschke et al., 2003) and water ice particles (Tomsic et al., 2003) of sizes $r_d \gtrsim 10$ nm fragment to a high degree. For water ice, molecular dynamics simulations and experimental evidence show a dependence of impact angle on the degree of fragmentation (Tomsic et al., 2001). Moreover, bulk properties can be used as a good approximation for those sizes. The case for smaller particles approaching the sub-nano scale is somewhat more complicated, as the yield stress, cohesive energy and work function can change as one approaches the atomic size limit (Qi and

Wang, 2002; Rennecke and Weber, 2014). We motivate our choice of fragment size limits in section 3.1. In the current work we employ two slightly different fragmentation models for pure metal particles and ice particles with impurities of meteoric smoke. However, it should be noted that the size distribution of fragments have the same proportionality with size in the two models, and thus the application of our model in the two cases have certain resemblance.

Figure 1 shows a sketch of the impact geometry. For the impact energies encountered in this paper, collisions can be assumed

to be fully plastic (Rennecke and Weber, 2014), and we thus can utilize Hertzian deformation theory for the main projectile of radius $R$. The contact area over which a capacitive coupling is established (as described in section 2.2 below), is defined as (Wang and John, 1988):

$$A = \alpha \pi r_p^2 \tag{1}$$

$$\alpha = \left( \frac{5}{4} \pi^2 \rho_p v_p^2 [k_t + k_p] \right)^{\frac{2}{5}} \tag{2}$$

where $r_p$ is the projectile radius, $v_p$ is the projectile impact velocity, $\rho_p$ its density and $k_t$ and $k_p$ the target and projectile elasticities defined by the Young's modulus $E$ according to $k_i \approx 0.89/\pi E_i$. We assume that impact duration ($\gtrsim$ picoseconds) is long enough to establish charge equilibrium. Considering the thermal speed of electrons, their mobility is by far high enough to obtain equilibrium for temperatures during impact. Furthermore, we have defined a parameter $h$ which gives the height of a cylinder of cross-section $A$ in which all the material is fragmented. Molecular dynamics studies (Tomsic et al., 2003) and rocket

results for low impact velocities (presented below) suggest that part of the grain material is decoupled from the rest. Thus only a small part of the projectile contribute to the capacitively generated charge. Such an understanding implies in our model, for velocities well below the volume ionization regime, that only a fraction $3h\alpha/4r_p$ of the original particle is involved in charge production. The rest of the particle is shielded or decoupled from the target surface. As shown in our results below, $h \sim 0.1 r_p$ offers a good fit for rocket data. Note that we have disregarded polarization effects, as the characteristic polarization potential

switching times for particles of sizes used here, are likely much longer than the collision time (or contact time) (Havnes and Hartquist, 2016).





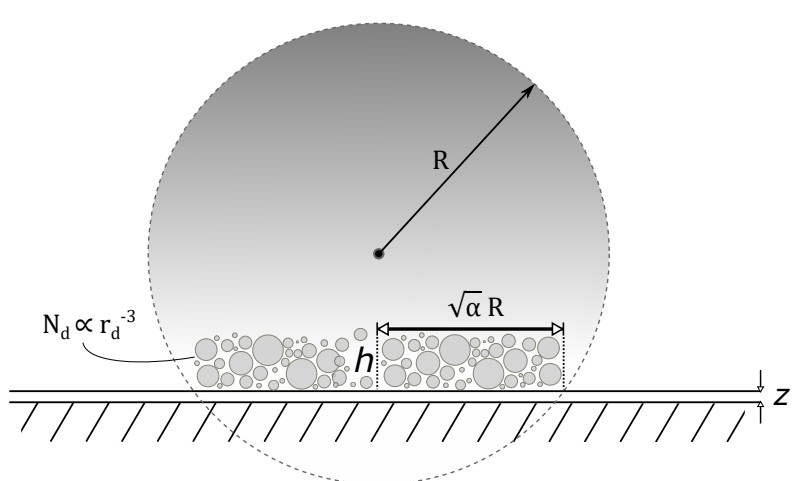

**Figure 1.** Contact geometry for the charging model of capacitive contact charging. The *d* subscript denotes *dust* here.

We employ the same parameterization of fragment size distribution in both the fragmentation-at-impact model (iron particles) and fragments-in-projectile model (ice particles containing meteoric smoke particles – MSPs), namely:

$$N_f(r_d) = N_0 r_f^{-3} \tag{3}$$

where $r_f$ is the fragment radius, $N_0$ is a constant defined by the available volume for fragmentation. This similarity arises from the fact that grain-grain or grain-surface collisions (Evans, 1994) and condensation of dust (Antonsen, 2019) yields the same $r^{-3}$-dependency. The grain-surface collision may be viewed as a collision cascade in free space, which typically yields similar size dependencies. It has also been confirmed that fragments of mesospheric ice containing impurities of meteoric smoke have a size distribution consistent with this model (Antonsen et al., 2017). Motivated by Antonsen and Havnes (2015), we also assume that for ice containing MSPs, the ice will evaporate quickly and moreover have a much lower affinity for charge exchange. Thus it is only the embedded MSPs that contribute to the charge production. It should be pointed out that the two fragmentation models are fundamentally different in that for iron particles the fragments are produced at impact, while for ice particles the fragments (MSPs) keep their original size distribution resulting from condensation and/or coagulation. This latter model is consistent with findings on how ice particles containing impurity fragments are detected in sounding rocket impact probes at speeds $\sim 1000$ ms$^{-1}$. Throughout the discussion below, we assume that the fragments are spherical grains. We also take into account lattice sphere packing, and use a value of $70\%$; representable for HCP, FCC, CCP and BCC lattice structures. The fragment size distributions are binned with a bin width of 0.01 nanometers and use a default size range of 0.5 to 4 nanometers in our numerical simulations. For some materials, such as iron, the lower size limit must be shifted in order to disregard quantum effects, as discussed in 3.The size distribution of fragments in a complete fragmentation of a 30 nanometer ice particle with embedded MSPs is shown in figure 2. The produced charge $Z_{tot}$ is calculated with our capacitive contact




charging model. The value of $Z_{tot}$ is a factor $\sim 5 - 10$ larger than what is usually measured with sounding rockets (Havnes et al., 2014), and is thus a confirmation that only part of the projectile particle contributes to the measured charge. We elaborate on the charge yield scaling for fragmented particles in Appendix A.

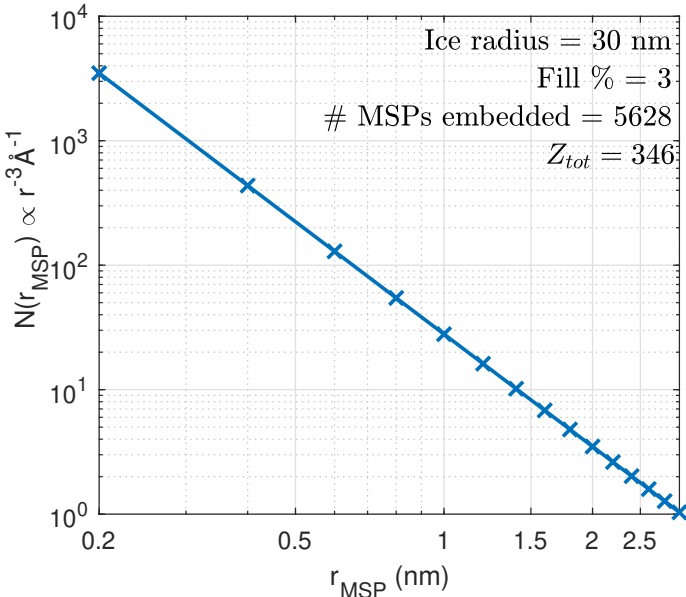

**Figure 2.** Size distribution of parameterizerd MSP particles inside an ice particle of size 30 nm. Note that $Z_{tot}$ is the upper bound on charge production for the case of complete charging of fragments. Annotated values show the result of employing our charging model on such a grain.





## 2.2 Contact Charging

The motivation for the current work is that at low impact speeds $\lesssim 1000 \text{ ms}^{-1}$ the charging of dust can be dominated by a capacitive charging mechanism where the projectile particle and (metal) target surface has an effective co-capacitance dependent on the difference in work function between target and projectile. As described in section 2.1 above, we take into account that

incoming projectile particles fragment at the speeds investigated, and we present a theory of contact charging for the individual fragments in the current section. The tensile strength of fragments increase with decreasing radius, and we assume that a capacitive charging model utilized on *fragments* is valid for much higher speeds compared to when utilized on single projectiles. In the following we give a short introduction to the contact charging theory presented by Wang and John (1988) applied on our problem. It is assumed that all impacts arrive perpendicular to the target, i.e. an impact angle $\theta = \pi/2$.

The fundamental mechanism behind contact charging as investigated here, is a capacitive coupling between a particle and a surface over an effective separation $Z_e \sim 10^{-9}$ m (Dahneke, 1972). The produced charge can be described as a function of time by:

$$Q = CV_c \left(1 - e^{-\frac{\Delta t}{\tau}}\right) \tag{4}$$

where $V_c$ is the difference in work function of the contacting materials, $C = \epsilon A Z_e$ the capacitance and $\tau$ the charge relaxation

time. For conductors, $\tau \sim 10^{-17} - 10^{-19}$ s, and $1 - \exp(-\Delta t/\tau) \approx \Delta t/\tau$, and it can be shown that eq. (4) reduces to:

$$Q = \chi \pi r_p^2 v_p \frac{\epsilon V_c}{Z_e} \left(\frac{4\rho_p}{3Y}\right)^{\frac{1}{2}} \tag{5}$$

where we have introduced the yield stress $Y$ of the material that yields first and the permittivity $\epsilon \approx \epsilon_0$. The parameter $\chi$ is a constant between 0 and 1 that we have introduced in our parameterization of fragment charging. It can be understood as the proportion of fragments which are charged until equilibrium, or alternatively as the charging probability of a single fragment.

The $\chi$-parameter is thus in essence a fitting parameter, and we note that we do not include the possibility of size dependency of it in our model. Note that it was used (set to non-unity) when producing the results in figures 4 and 5. In table 1 we summarize the parameters used to produced the presented results. It must be noted, that even if bulk values can be extrapolated to very small particle sizes, that some of the parameters utilized in the numerical computations here vary with temperature which we do not take into account here: The static, i.e. low frequency equilibrium, relative permittivity of ice increases with decrasing

temperature (Auty and Cole, 1952; MacDowell and Vega, 2010). The Yield stress and Young's moduli for olivine and ice may also change significantly with temperature (Evans and Goetze, 1979; Nunez-Valdez et al., 2010; Nimmo, 2004). While metals are somewhat more resilient to changes in parameters related to internal stress, their work function increases with decreasing size due to a change in polarizability as particles become small (Wood, 1981).





**Table 1.** Material properties used in the calculation of contact charging yields for silver (Ag), iron (Fe), water ice, Stainless steel (SLS) and a metoric smoke analogue (MSP).

|  | Ag | Fe | Ice | SLS[†] | MSP[††] |
|---|---|---|---|---|---|
| $\rho[\text{kgm}^{-3}]$ | 10500 | 7874 | 980 | 7800 | 3000 |
| $E[\times 10^{9}\text{Nm}^{-2}]$ | 104 | 150 | 9 | 170 | 200 |
| $Y[\times 10^{6}\text{Nm}^{-2}]$ | 330 | 50 | 10 | 50 | 50 |
| $\phi[\text{eV}]$ | 4.7 | 4.5 | – | 4.4 | 7.3–8.5 |

[†] Stainless Steel 316. [††] Meteoric Smoke Particles; here Olivine at 300 K (Rapp et al., 2012).

## 2.3 Shock Wave Impact Ionization

It is recognized that impact ionization is a combination of mechanisms, each dominating for certain parts in a wide velocity range. In this paper, we take it as fact that the impact ionization will tend towards a volume ionization mechanism – as a consequence of a Thomas-Fermi model for electronic structure – as impact speeds exceed $\sim 50\,\text{kms}^{-1}$ (see e.g. Auer (2012)).
For velocities below such high speeds, the accepted and most widely used model for impact ionization is that of Drapatz and Michel (1974), which describes ionization as a result of shock waves propagating through colliding entities.

The model of shock wave ionization does not, however, describe ionization for the entire velocity range below $\sim 50\,\text{kms}^{-1}$. This was also recognized in the earliest formulations of the theory and discussions on its validity. There are in fact at least two velocity regimes – low ($v_p \lesssim 5 - 10\,\text{kms}^{-1}$) and high ($v_p \gtrsim 10\,\text{kms}^{-1}$) – that display different semi-empirical charge yields. The charge production mechanism is different in the two velocity regimes, and one focus of the present work is to put the theory of shock wave ionization in the low velocity regime under scrutiny. In section 4.1 we discuss in detail the thermodynamics of the low velocity regime shock wave ionization as it was formulated by Drapatz and Michel (1974).

For the high velocity regime, the shock wave ionization model assumes that the ionization state freezes at some point during expansion of the gas ball arising from impact (Raizer, 1960; Kuznetsov and Raizer, 1965). The degree of ionization can then be calculated from a Saha-equation (Dresser, 1968):

$$\frac{n_+ n_-}{n_0} = \frac{2\psi_+}{\psi_0}\left(\frac{2\pi m k_B T}{h^2}\right)^{\frac{3}{2}}\exp\left[-\frac{eV_I^1}{k_B T}\right] \tag{6}$$

where $n_+, n_-$ and $n_0$ are the respective number densities of ions, electrons and neutrals; $\psi$ are atomic weights for the ionic and atomic states; $T$ denotes the temperature; $k_B$ denotes Boltzmann's constant; $|e| = 1.6 \times 10^{-19}$; $h$ denotes Planck's constant, and $V_I^1$ denotes the first ionization potential of projectile atoms.

For the low velocity regime, Drapatz and Michel (1974) pointed out that impurity ionization predominantly from alkali metals in the projectiles was responsible for the charge yield. One must then ulitize that electrons bound in metals follow a distribution (Copley and Phipps (1935); for potassium on tungsten):

$$n_- = 2\left(\frac{2\pi m k_B T}{h^2}\right)^{\frac{3}{2}}\exp\left[-\frac{e\phi}{k_B T}\right] \tag{7}$$





where $\phi$ is the work function of the projectile. This ultimate leads, after insertion into eq. (6), to what can be recognized as the Saha-Langmuir equation:

$$\frac{n_+}{n_-} = \frac{\psi_+}{\psi_0} \exp\left[-e\frac{\phi - V_I^1}{k_B T}\right] \tag{8}$$

where for the materials discussed here $\psi_+/\psi_0 \approx 2$.

In the following, we consider the situation that both target and projectile are conductive. This means, in simplified terms, that electrons can easily move between potential wells on the surface of the projectile and target and have time to equilibrate the charge within the collision time $\tau$. Then, in all generality, the charge yield can be described by power laws in both velocity and particle size. The charge production is thus often described by the formula:

$$Q[\mathrm{C}] = \gamma m_p[\mathrm{kg}]^\alpha v_p[\mathrm{kms}^{-1}]^\beta \tag{9}$$

where the constants $\gamma, \alpha$ and $\beta$ are all strongly dependent on material properties and velocity regime (see e.g. Mocker et al. (2013); Collette et al. (2014); Kissel and Krueger (1987)). For the application on micrometeoroid impacts on spacecraft, the velocity exponent most widely cited value for velocity dependence $\beta \approx 3.5$ adopted from McBride and McDonnell (1999), however values of $2.5 - 6.2$ have been reported for common spacecraft materials (Mann et al., 2019). The exponent $\alpha$ for mass dependence is usually found to be $\sim 0.7$ for the low velocity regime and close to unity for the high velocity regime. It has

already been pointed out, by Kissel and Krueger (1987), that for low velocity impacts $(v_p < 5\,\mathrm{kms}^{-1})$ $\alpha$ should be close to $2/3$. This is to say that the charge yield is propotional to the incoming projectile cross section $r_p^2$. The same authors also pointed out that, both in the low and high velocity regime, ionization at the target could also be described by the same power law stated in eq. (9). However, the exponent for mass dependency would be one dimension in particle size lower; i.e. $Q_{\mathrm{target}} \propto r_p^2$ for high projectile velocities and $Q_{\mathrm{target}} \propto r_p$ for low velocities. In this work, we do not discuss the additional effect of direct target

ionization. In section 4.1 we elaborate in that low velocity impact charging must probably be described by a different physical mechanism than shock wave ionization as desribed here.

## 3   Results

In sections 3.2 and 3.3 below, we present results from calculations of contact charging for projectiles of metal and projectiles of ice on metal surfaces, respectively. The simulations employ the fragmentation and charging models described above. For

calculations of ice-on-metal charge yields, we assume for that the ice particles are contaminated with meteoric smoke particles. This model is descriptive for icy dust particles (or aerosols) in the Earth's mesosphere, and we compare our results with in-situ measurements of mesospheric ice.

### 3.1   A note on projectile size and fragment size sensitivity

The choice of default projectile grain size (30 nm) in the presented model results below, may be motivated by that it is a typical

size of mesospheric icy dust grains usually encountered by sounding rockets. It is also among the smaller projectile sizes (see



e.g. figure 2 in Mann et al. (2019)) which can be readily generated in typical dust accelerator experiments. Moreover, the charge produced at a specific projectile size, can be directly scaled to larger sizes according to the scaling relation given in eq. (A5). Normalizing the calculated yield to the projectile mass furthermore allows for direct comparison to semi-empirical laboratory results.

We must also address the choice of limits in fragment size distributions, and sensitivity to changes in the lower cut-off limit. In their treatment of collisional charging of interstellar grains, Draine and Sutin (1987) argued on the basis of results by Omont (1986), that bulk properties would sufficiently describe grains of PAHs down to sizes of only 3 Å – or $\gtrsim$ 30 molecules. That is to say that, at least for carbonaceous or PAH dust, one can model particles as conducting spheres and disregard quantum effects when calculating the equilibrium charge due to polarization (image) and capture of charged species. We use this as one reason

to model fragments of impacted projectiles as conducting spheres down to sub-nanoscale. In our framework of modelling metal on metal interaction, we must note that the atomic interspacing is $\sim$ 2 Å in a BCC lattice structure. Thus, to obtain a grain that satisfies the constraints used in the references above, we must increase our cut-off to $\sim$ 6 Å for iron fragments. This is also in agreement with the findings of Jones et al. (1996) for minimum fragment size in low velocity impacts. In figure 3 we compare the response of our contact charging model for different low-size cut-offs in the fragment size distributions. The differences

are small – an increase from 2 Å to 8 Å in cut-off only decreases yield by around 30%. Thus we increase the low-size cut-off of iron fragments to 7 Å in this work while still keeping the conducting sphere assumption elaborated on above.

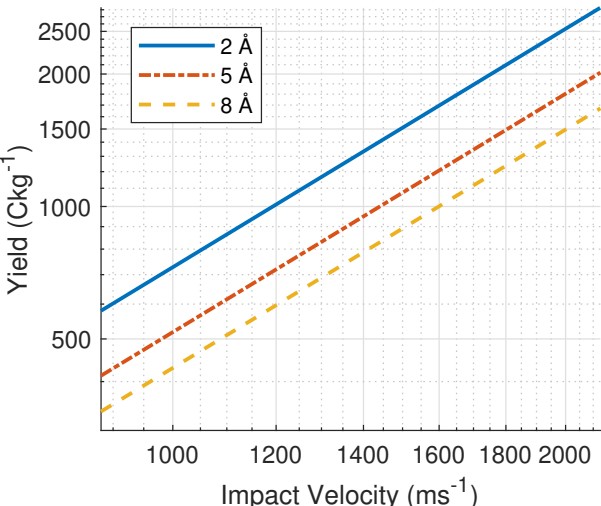

**Figure 3.** Sensitivity of contact charge generation in an Fe-on-Ag collision to different values of lowest allowed fragment sizes. Cut-off values are labelled in the legend.





## 3.2 Metal–metal collisions

In the following simulations, we have used iron as projectile material and silver as target material. This is due to that experiments with this combination have been done in the past both at the LASP dust accelerator (Collette et al., 2014) and at the Heidelberg dust accelerator facility (Mocker et al., 2013). The Fe–Ag combination may be applicable to dust impacts on space-

craft as iron is common in e.g. micrometeorites. It was also used as an example by Drapatz and Michel (1974) in the original formulation of the shock wave ionization theory. Thus, it is possible to compare our results with several others. It should be noted, however, that different experiments may have different complex geometry and working principles. The data produced by the different experiments may therefore have intrinsically systematic differences and direct intercomparison must be done with care.

Figure 4 shows a comparison of our calculations of iron projectiles on a silver target to the semi-empirical results obtained by Mocker et al. (2013). Their data (blue curves) show a discontinuity – or sudden increase in yield – at $\sim 11$ kms$^{-1}$. Since the thermodynamics and chemistry of the fragmentation process might not be valid at speeds much higher that this limit, we focus on comparing our curves with the experimental data at speeds below the discontinuity. The lower limit of contact charging, showed as a solid black line, is the case where we consider that the entire original projectile participates in the capacitive

coupling. This assumption implies that the entire particle must be bound together, while it still allows for plastic deformation. Such a situation is not probable, but provides a lower boundary on the charge production. The fragmentation model results is sketched as a dashed black line. In general, it has a slope very close to the semi-empirical model, but its values are to two orders of magnitude higher. We note that figure 4 shows a default run with default parameters suitable for bulk material, and refinement of these will lead to different results, as shown below. Moreover, as shown in Appendix A, there is a strong

dependency on the parameterization of the fragmentation size distribution. Refining the parameterization yields a much better similarity to the experimental results. Although it is not the purpose or motivation of this work to explain the entire charging mechanism at low impact speeds with fragmentational contact charging, we nevertheless have calculated a best fit of our model to experimental data with reasonable pararameters. In figure 5 we show the result of a simultation with a set of parameters that produces a 'best fit'. The yield stress was increased by a factor three compared to figure 4. The yield stress is in any case a

parameter with a significant uncertainty for nano- and microscale particles. Moreover, the fraction of fragments that become charged was reduced to $\chi = 1\%$. This parameter is difficult to define, and has a large intrinsic uncertainty. A final adjustment was made to the size distribution; where the smallest possible fragment size was changed from the default value of 0.5 nm to 0.7 nm. Since many of the parameters used in our charge model are valid for bulk projectiles, the validity of extrapolating the model to sizes $\lesssim 10$ nm can be a topic for discussion. In a more rigorous treatment, one may have to take into account

curvature and polarization effects for the smallest fragments. Nevertheless, our model shows that at low speeds, fragments can indeed produce charge in a capacitive coupling very efficiently as opposed to ionization through a Saha-process. We also leave the untreated issue of how much the pre-charge, which can be very large for the large projectiles that dominate the low velocity range, contributes to the yield.



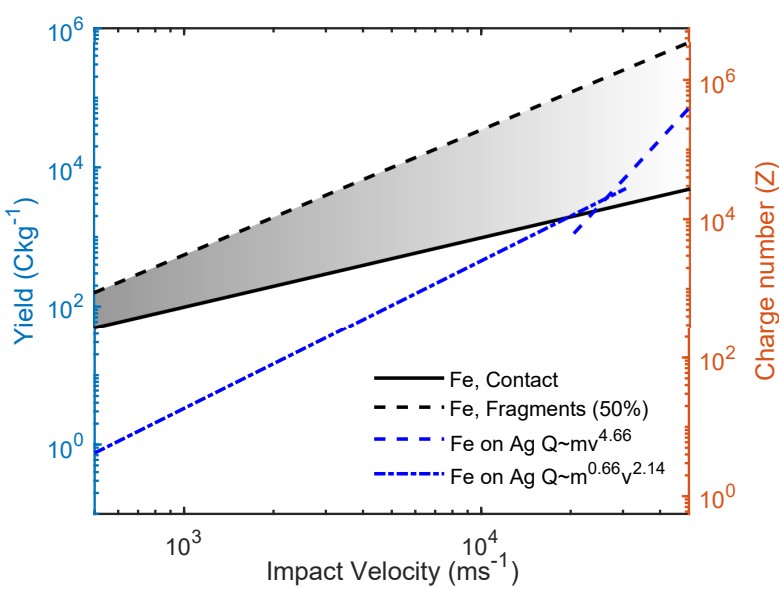

**Figure 4.** Simulation of contact charging of iron projectiles ($r_{p,0} = 30$ nm) on a silver target with (black dashed line) and without (black solid line) a fragmentation model. The power laws plotted in blue were aquired experimentally by Mocker et al. (2013). In this calculation, the yield pressure of iron was set to $Y = 50$ GPa, and the fragment size span was set to $[0.2, 3]$ nanometers. The label '50%' indicates that in this calculation, $\chi = 0.5$ cf. equation (5).

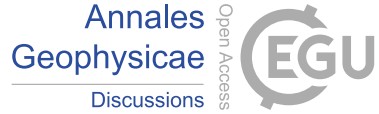



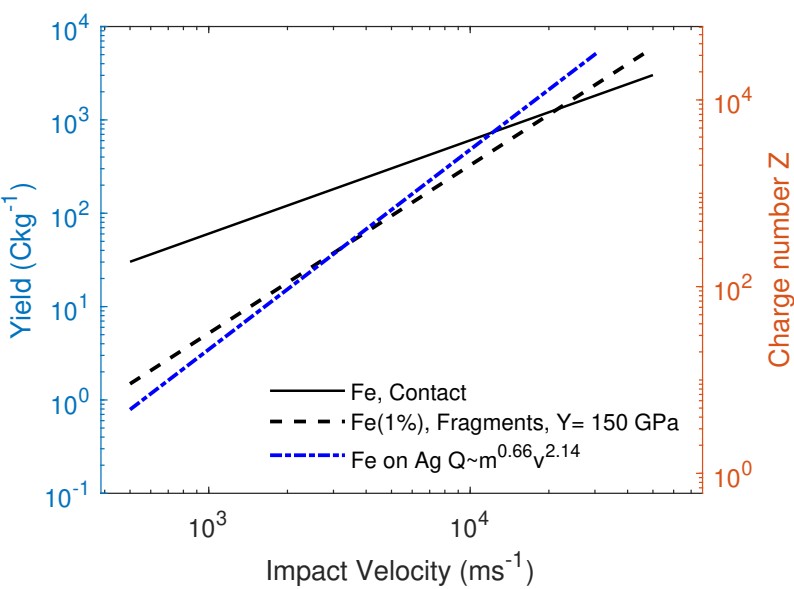

**Figure 5.** Simulation of contact charging of iron projectiles on a silver target. This simulation was a "best fit" run, in which the fraction of charged fragments, yield pressure, and fragment size span was allowed to change. This shows the case for $\chi = 0.01$ cf. equation (5), $Y = 150$ GPa and fragment size span [0.7, 3] nanometers.





### 3.3 Ice–metal collisions

It is technically challenging to set up laboratory experiments for studying low impact velocities ($\lesssim 1$ kms$^{-1}$) and small projectiles ($\lesssim 100$ nm) simultaneously. Dust accelerators typically use samples of projectile particles which span several orders of magnitude, while the energy is fixed and determined by the strength of a static accelerating potential (see e.g. Thomas et al.

(2017) for LASP setup). Such a configuration implies that only very large projectiles will have low impact speeds.

It is however possible to use sounding rockets to obtain a point measurement in the low size and speed range: Typical sounding rockets utilized in upper atmosphere research operate at low speeds $\sim 1$ kms$^{-1}$, and naturally occuring dust particles in the mesosphere ($\sim 50 - 100$ km above sea level) typically have sizes $\sim 1 - 100$ nm. Thus, in-situ measurement can be compared to laboratory measurements for certain experimental setups. For the results below, we utilize the rocketborne Faraday

impact probe MUDD (MUltiple Dust Detector). Inside it, incoming projectile dust particles hit a slanted stainless steel plane on which they deposit pre-charge and produce contact charge. A detailed technical description, projectile dust dynamics and utilization of the instrument can be found in Havnes et al. (2014); Antonsen and Havnes (2015); Antonsen et al. (2017).

As previously stated, we utilize that dust grains in the mesosphere are contaminated with meteoric smoke – recondensed and agglomerated remnants of meteoric ablation. In figure 6 we show the result of two limiting cases of contact charging of

'dirty ice'. The solid black line decribes a situation where no impurities contribute to the produced contact charge (the pre-charge is assumed to be zero). This might be plausible for very low speeds, where ice particles of sizes $\sim 10$ nm experience less fragmentation (see e.g. Tomsic et al. (2003)). For projectile speeds on the order of a few hundred meters per second, the fragmentation model (dashed line) should provide a more physically sound charge yield. For a typical rocket speed of 800-1000 ms$^{-1}$ the fragmentation model and single projectile have roughly the same yield. We find that the predicted charge number

for this velocity range is consistent with what has been measured with rocketborne Faraday cups (Havnes and Næsheim, 2007; Havnes et al., 2014). The gray shaded area shows a values of the predicted charge yield where the ice particles have a capacitive coupling, but are allowed to have a non-unity dielectric constant (cf. Wang and John (1988)). This effectively means that the ice particles are insulating, which may be a better description. The true yield of a pure ice projectile should therefore probably lie below the solid black line in figure 6.

In the following we attempt to simulate the current recorded by MUDD during a flight in the MAXIDUSTY campaign (Andøya Space Center, June 30th 2016). We assume the finding of Antonsen (2019); that small ice fragments thermalize and evaporate very quickly and MSP fragments dominate the produced signal. The size distribution of MSPs inside ice was previously found to be $N \propto r_f^{-[2.6,4.4]}$ (Antonsen et al., 2017), thus an exponent of $-3$ should fit well. Consequently, we can employ the same fragmentation model as for metal-metal collisions. We use a volume content of MSPs of 1%, which is in the

middle of the range of what has been found from rocket measurements and satellite measurements (Hervig et al., 2012). Other parameters used in the calculations are listed in table 1. We also note, that the impact plane in MUDD is slanted, however, we assume that the contact time is on the same order as for head-on collisions and long enough to reach equilibrium.

The rocket traversed a dust layer situated at $\sim 81 - 87$ km at a velocity $\approx 810$ ms$^{-1}$. Accurate number densities and sizes of ice particles were found by a combination of ALOMAR RMR lidar data, in-situ photometer and DUSTY Faraday cup data



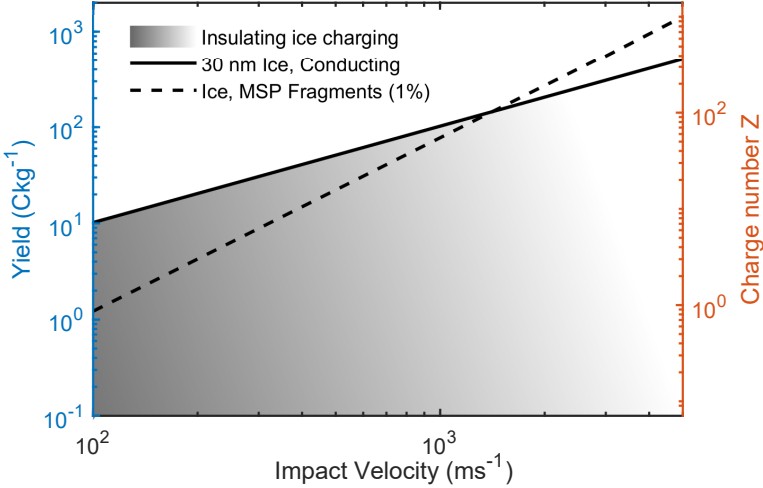

**Figure 6.** Contact charge yield of 30 nm ice particles with MSP impurities impacting on stainless steel. The shaded area shows possible yields for the case of a mixture of insulating and conducting particles.

as described in Havnes et al. (2019). The electron and ion density acquisition for this flight is explained in the same reference. Figure 7 shows the comparison between measured MUDD currents and simulated MUDD currents using the described fragmentation model. The two curves display a very high similarity down to the smallest scales, and only differ significantly at the upper ∼ 1 km of the dust layer. Combined with the fact that the fragments – i.e. the embedded MSPs – do not carry significant pre-charge, our results presents a convincing case for contact charging being the dominant charging process for the speeds and particle sizes encountered when probing the dust in the Earth's mesosphere.



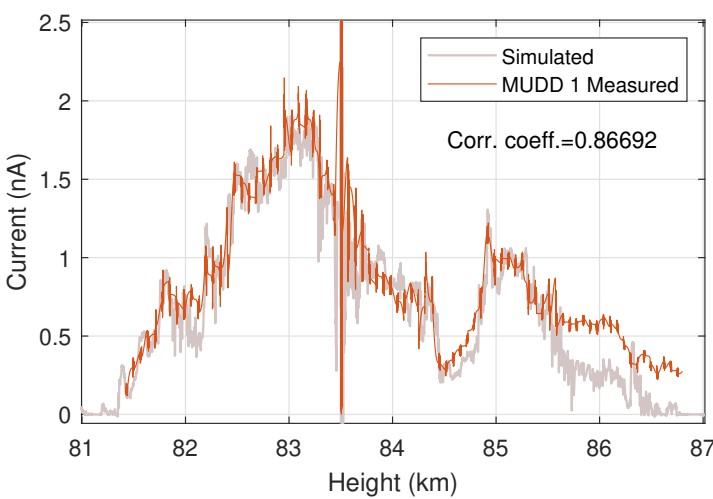

**Figure 7.** Measurements from the impact Faraday cup MUDD flown on the MXD-1 sounding rocket payload (red) and a best fit from simulation of contact charging (grey) using the fragmentation model descibed in 2.1. $V_c = 0.5$ eV was found to be the best fit for fragments of density $\rho = 3000$ kgm$^{-3}$ and yield pressure 50 MPa. The minimum fragment size threshold was set to 0.3 nm.





## 4 Discussion

As presented in section 2.3, it was pointed out by Kissel and Krueger (1987) that the Saha-Langmuir solution (SLS) from Drapatz and Michel (1974) underestimates impact charge generation for speeds $\lesssim 5$ kms$^{-1}$. The low velocity solution of their theory assumes that charge is generated mainly by impurity diffusion through the molten projectile material. Although Mocker et al. (2013), whose results we have used for comparison, conclude with an agreement with the SLS, it must be noted that they find that the apperance of Fe in impact time-of-flight mass spectra occurs at much lower speeds (3.6 kms$^{-1}$) than the SLS predicts. Hydrocode simulations aiming at predicting the threshold of impact plasma generation in iron-on-metal collisions have found a threshold of 8 kms$^{-1}$ (Ratcliff et al., 1997). The takeaway from these notions, is that both direct impact plasma generation and the SLS probably is insufficient in explaning charge generation in low velocity impacts.

To investigate the applicability of an SLS at speeds on the order of 1 kms$^{-1}$, we look closer at the thermodynamics of the process in section 4.1. In section 4.2 we discuss possible areas of application of our contact charging model, with emphasis on spacecraft and sounding rocket observations.

### 4.1 Thermodynamics of Low Velocity Limit of Shock Wave Ionization

A first order estimate of the mean diffusion distance of an impurity ion inside a cooling – i.e. solidifying – metal grain, can be found by recognizing that the diffused area must be $D(T)\tau_s$, where $D(T)$ is the temperature dependent diffusivity over a solidification time $\tau_s$. We have that the one-dimensional mean diffusion distance is

$$\delta r = |(D(T)\tau_s)^{\frac{1}{2}}| \qquad (10)$$

This can be interpreted as the thickness of a shell from which impurities can reach the surface of a cooling grain. For the purpose of comparing diffusion of alkali impurities through iron particles with the results of Drapatz and Michel (1974), we utilize the mean diffusivity $\overline{D(T)} = 5 \cdot 10^{-9} \exp(-5000/\overline{T})$ m$^2$s$^{-1}$.

The available amount of substance from which ions can be released from a grain of radius $r_f$ has a volume of – assuming the particle has bulk properties, which is suitable for particles on the order of $\sim 10$ nm:

$$\delta V(\tau_s) = \frac{4\pi}{3}\left[r_f^3 - (r_f - \delta r)^3\right]. \qquad (11)$$

Then the upper bound on the number of impurity ions released from a single grain becomes:

$$N_{im} = \mathrm{K}\frac{4\pi\rho_p\xi}{3M_p}\left[r_f^3 - (r_f - \delta r)^3\right] \qquad (12)$$

where $\xi$ is the impurity content by volume and $\rho_p$ and $M_p$ are the mass density and molecular mass of the grain ('projectile') material respectively. We use here that $\xi \sim 1\%$, which is representable for alkali metal content in Earth's crust and in raw smelted iron and steel. K is the atomic packing factor, which is set to 0.7 in our calculations. The resulting single charge impurity ionization predicted by the Saha-Langmuir equation then becomes

$$Z_{im} = N_{im}\frac{n_{im}}{n_{im} + n_{Fe}} \approx 2N_{im}e^{-\frac{e\phi - V_I^1}{k_BT}} \qquad (13)$$





where $e$ is the elementary charge, $\phi$ is the work function of the impurity material and $V_I^1$ is the first ionization potential. The difference between these for potassium (K), which is used in this work as a dominant impurity, is $\phi - V_I^1 \approx 1.8$ eV. The factor 2 arises from the statistical weights in equation (8).

It is clear that we also require a parameterization of the temperature of inside the expanding shock. For this purpose we utilize that the relationship between the shock front velocity $u$ and the projectile velocity $v_p$ is governed to first order by the ratio of the difference in mass density between the projectile and target through energy conservation. We have for specific energy

$$\varepsilon = \frac{u^2}{2} = \frac{v_p^2}{2} \left/ \left( \frac{\rho_p}{\rho_t} \right)^{\frac{1}{2}} + 1 \right. \tag{14}$$

which for our example case of iron projectiles on a silver surface yields $u \approx 0.62 v_p$. This is to say that $\sim 40\%$ of the initial energy goes into expansion of the shock. It can be shown than the temperature behind the shock for monoatomic gases ($\gamma = 5/3$) is (Zel'Dovich and Raizer, 1967):

$$\frac{T}{T_0} = \frac{5}{16} \mathrm{Ma}^2 \tag{15}$$

where $\mathrm{Ma} = u/v_{th,N}$ is the shock front Mach-number and $T_0$ is the pre-collision temperature.

In the following calculations of impurity ionization production, we have used that the solidification temperature of nanoscale iron particles is 1000 K (Fedorov et al., 2017). Furthermore, we employ a cooling rate of $10^{12}$ Ks$^{-1}$, which has been found from molecular dynamics simulations to be representable for nanoscale metal particles (Shibuta and Suzuki, 2011). This gives typical solidification times of $\tau_s \sim 10^{-9}$ seconds, which is two orders of magnitude smaller than the cooling time used by Drapatz and Michel (1974). In the current model, we restrain the diffusivity coefficient with a hard stop at the solidification temperature, and do not parameterize solidification/crystallization effects. Moreover, we do not discuss here the evaporation of impurities from the surface of the main particle; we simply assume all impurities are removed, and thus present an upper bound on impurity charge production. By the set of equations above, we find that the limit for impurity production for a 3 nm iron grain is a Mach-number of 3.3, corresponding to a velocity $v_p \sim 1$ kms$^{-1}$.

In figure 8 we have not parameterized the temperature decrease in the expanding gas volume behind the shock. However, depending on whether or not thermodynamic equilibrium can be reached or not, this effect might be a significant inhibitor of thermal ionization described by the S-L equation. For an adiabatic expansion, we have that

$$\frac{T(\tau_s)}{T_0} = \left( \frac{V_0}{V(\tau_s)} \right)^{\gamma-1}. \tag{16}$$

Thus, in the case of a 30 nm projectile particle and a solidification time of $\tau_s = 10^{-9}$ seconds, it is found that the limiting expansion velocity to accomodate diffusion should be on the order of $u \sim 100$ ms$^{-1}$, which is clearly never the case. In this regard, we note that the assumption of thermodynamic equilibrium may not be suitable for the set of parameters encountered in the current work. Moreover, we must note that the emissivity of nanoscale dust grains is strongly dependent on size and material properties, so the cooling time may also need refinement (Rizk et al., 1991).

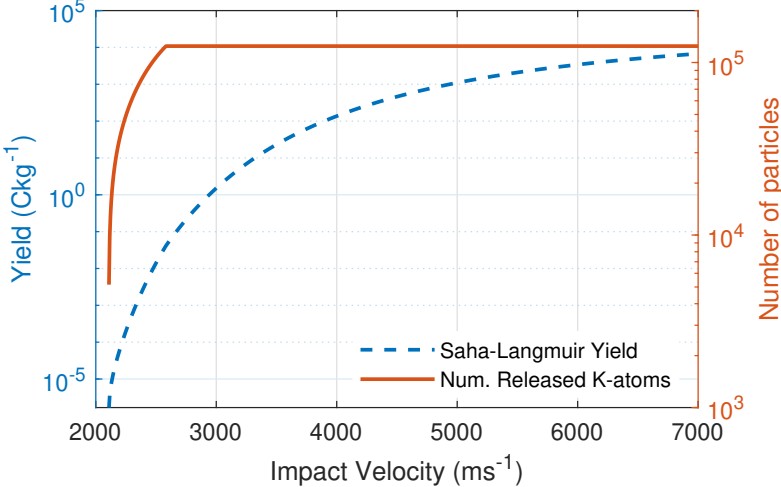

**Figure 8.** Results from calculation of impurity (1% Potassium) charging using the Saha-Langmuir equation (blue, dashed) and fragmentation model described in this work. The solid red line show the number of released K-atoms as a function of velocity, and therefore constitutes a theoretical upper bound on the charge number (for singly charged ions).

We summarize our result of low velocity impact charging in figure 9. The SLS is vanishing below $\sim 2\ \mathrm{kms}^{-1}$. The contact charge solution overestimates the experimentally acquired yield in the entire range. The two curves have almost the velocity and mass dependence, and possible downshifts of the contact charge solution were discussed in section 3.2. Notably, we have used $\chi = 1$ here, which provides an upper boundary on the charge yield. We must note that in the mechanism proposed in the

current work, we have not taken into account charge production at the surface. Such an effect may be expected to scale with the particle radius rather than cross-section at low speeds, as mentioned in section 2.3. To first order, one may therefore disregard such additional charging, as is introduces another layer of complexity into modelling efforts. There is also a possibility of impurities on the surface – which are arguably always present in metals – that can produce additional charging. Extending our theory to include the contributions from impurities may require a treatment of surface chemistry and evaporation gas

microphysics which is beyond the goal of our study. Nevertheless, our results must always be read with the ulterior notion that volatile impurities such as alkali metals can introduce additional charge.

## 4.2   Relevance for Dust Detection on Spacecraft and Rockets

In section 3.3 we demonstrated the applicability of our model to rocket measurements of dust (or aerosols) in the upper mesosphere of Earth. Other related types of dust, namely those originating in the ablation of meteors in the altitude range $\sim 70$

to 140 km, are candidates for comparison with a contact charging model. Free dust grains of meteoric origin have recently been observed by sounding rockets (Havnes et al., 2018), and other novel experiments have been aimed at investigating such particles (see e.g. Strelnikov et al. (2018)).



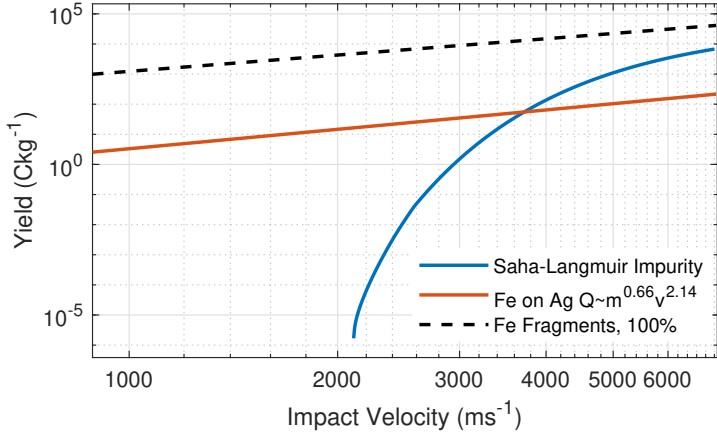

**Figure 9.** Comparison of specific yields from our contact charging model (dashed) to the Saha-Langmuir solution from Drapatz and Michel (1974) (solid blue) and the semi-emprical law obtained by Mocker et al. (2013) (solid red). All lines represent Fe-on-Ag impacts, and in the Saha-Langmuir solution we have used potassium as the impurity. The label '100%' in the legend corresponds to $\chi = 1$ and indicates that all produced fragments collide with the surface.

The number of *catalogued* man made space debris objects in the near-Earth space is already on the order of $10^4$, with the number of objects increasing inversely with size (Klinkrad, 2010). Estimates of the number of objects smaller than 100 $\mu$m have large uncertainties, but have been cited as in the order of tens of billions (Schildknecht, 2007). The probability of any satellite encountering a space debris object during its lifetime is therefore nonvanishing. Besides this, typical orbital speeds of

Low Earth Orbiting satellites are $\lesssim 8$ kms$^{-1}$, and decreasing with increasing orbital altitude. Thus, even without considering whether the grains of debris are pro- or retrograde, debris-satellite interactions may readily occur at speeds relevant for contact charging. Additional charging or upsets due to contact charging is arguably undesirable for e.g. satellites measuring plasma parameters.

Based on recent observations by the Parker Solar Probe (PSP), Szalay et al. (2020) concluded that $\beta$-meteoroids ($\beta$ describing

the ratio of the radiation pressure force to gravity) dominate the recorded dust flux. They found that such grains typically have larger impact velocities than circularly bounded dust, but for $\beta \lesssim 0.5$, there may be a nonvanishing flux of $\beta$-meteoroids with impact speeds in the upper limit of the velocity range investigated in the present work for contact charging. From their results of modelling the dynamics of dust in bounded circular orbits, based on a model of Pokorný and Kuchner (2019), it is clear that there may well be a smaller number of impacts on PSP that can be traced to such dust grains.

Page et al. (2020) reported that although $\beta$-meteoroids can produce dust impact fluxes as measured on PSP near perihelion, some of the directionality in the dust flux data can be consistent with prograde circular orbit dust. At perihelion, the impact velocity of these is still to large to apply a contact charge model ($\sim 20$ kms$^{-1}$), however, it might be a possibility that one can use such a model on prograde dust further away from the Sun with lower impact velocities. In regards to further explorations





of the utilization of our model on spacecraft data, dependence of impact inclination in contact charge production should also be studied.

Another possible candidate for employment of our model on spacecraft data, is secondary ejecta. Secondary ejecta, which is material from craters generated by dust impacts on the spacecraft body, have energies much lower than the impacting grains.

Such secondary grains have been observed as stray light in optical images from e.g. STEREO (St. Cyr et al., 2009). Szalay et al. (2020) also noted that such secondary particles were observed with the WISPR experiment on PSP (see e.g. Vourlidas et al. (2016)), and that the ejecta correlated well with antenna measurements of dust impacts.

One impediment to utilizing our model on dust in space, is that it may be difficult to determine its composition and structure. In consequence, the work function of the projectile material may be unknown and moreover size dependent (Wood, 1981). In

some cases, where the projectile grains have very low conductivity, it may be required to either: 1. Assign an effective work function (Matsusaka et al., 2010) or 2. Extend our theory to insulating particles. The latter can in brief be described as letting the the the ratio $\Delta t/\tau \rightarrow 0$ in eq. (4). The result is a slightly lower velocity dependence in the charge production; $Q_c \propto v_p^{3/5}$. The charge production will also be significantly weaker than for conducting or semi-conducting grains (John et al., 1980; Wang and John, 1988).

The ESA Solar Orbiter (ESO) was launched in February 2020. Its orbit is different from PSP in that its perihelia are larger than $\sim 0.28$ AU throughout its lifetime – versus $\sim 0.046$ AU for PSP. In addition, Solar Orbiter has a planned 25 degree inclination in its nominal mission. The orbital parameters will ensure that ESO will encounter prospective bounded dust grains and $\beta$-meteoroids with generally lower velocities than for PSP. It is not in the scope of this paper to analyze the expected dust flux of ESO, but it can be expected that a larger number of dust impacts can involve a contact charging mechanism compared

to PSP.

## 5    Conclusions

In this work we have investigated the production of charge in impacts of projectiles of iron and agglomerates of ice and meteoric smoke on a metal surface at speeds $\lesssim 10$ kms$^{-1}$. We introduce a novel model of contact charging due to a capacitive coupling between metal surfaces and fragments of projectile grains. Here we show that our model is consistent with laboratory

measurements of Fe-on-Ag collisions as well as rocket measurements of icy dust particles on stainless steel. Our method can be utilized with a large range of projectile dust types, where the intrinsic properties of the grains are known. We also find that our theory may be used to explain certain observations of dust by the recently launched spacecraft NASA Parker Solar Probe and ESA Solar Orbiter. We moreover find that the currently accepted theory for impact charging at the speeds of interest here, namely the shock wave ionization theory of Drapatz and Michel (1974), is insufficient in explaining laboratory observations

of charge generation in metal-on-metal impacts. Consequently, we suggest that at low speeds, there must be a significant contribution to the produced charge by contact charging.





## Appendix A: Scaling relation for charge yield of fragmented particles

In this appendix we give a scaling relation for charge production by capacitive charging when employing a fragmentation model.

The available material from which fragments can form is given by the Hertzian deformation presented in section 2.1 above, and is:

$$V_c = \alpha h \pi r_p^2 \propto r_p^3 \sim Q_p. \tag{A1}$$

The largest possible spherical fragment (of volume $V_s$) that can be formed from this material, using $V_s(r_f) = V_c(r_p)$, has radius:

$$r_{\max} = \left( \frac{3h\alpha r_p^2}{4} \right)^{\frac{1}{3}} \propto r_p. \tag{A2}$$

As contact charging scales with the cross-section of fragments, we calculate the total surface area of all (discretely distributed) fragments:

$$S'_{tot} = \sum_{\forall i} \pi r_{f,i}^2 = \sum_{\forall i} \pi \left( N_0 r_{f,i}^{-3} \right) r_{f,i}^2 \tag{A3}$$

where we have used that the fragments are distributed in size according to $N_f \propto r_f^{-3}$. If we assume that the size distribution is continuous, we can moreover find that

$$S_{tot} = \pi N_0 \int_{r_{\min}}^{r_{\max}} \mathrm{d}r_f \cdot r_f^{-1} = \pi N_0 \ln \left( \frac{r_{\max}}{r_{\min}} \right) \tag{A4}$$

where $r_{\min}$ is the smallest possible fragment radius.

Now we recall the scaling $Q_p \sim r_p^3$ from eq. (A1), which constraints the amount of material available for fragmentation. Inserting the result from eq. (A2) into eq. (A4) we finally obtain that the charge production of a fragmented projectile particle scales with size (and velocity according to its dependence in $\alpha$):

$$Q_p \propto r_p^3 \ln \left( \frac{(3h\alpha r_p^2)^{1/3}}{4^{1/3} r_{\min}} \right) \propto r_p^3 \ln \left( \frac{r_p v_p^{4/15}}{r_{\min}} \right). \tag{A5}$$

This result is also intuitively reasonable; that since there are many more small particles than large ones, the surface area of the small particles contributes more to the total area and thus charge production. Moreover, we note that the sensitivity to the parameter $r_{\min}$ becomes even more important for size distributions with steeper inverse power laws than the one chosen here. This solution will never become unphysical (singularity as $r_{\min} \to 0$), as $r_{\min}$ has a natural lower bound. In this paper its dependence on velocity, which must be assumed that is has. However, we assume that that a feasible value would be on the order of $\sim 1$ Ångstrøm ($= 0.1$ nanometer), which is the order of the length of a single atom or molecule. As seen in section 3, our charge production model is relatively sensitive to this parameter.





*Acknowledgements.* This work was supported by the Research Council of Norway through grant number 262941. The rocket campaign and the construction of the rocket instrumentation was supported by grants from the Norwegian Space Centre (VIT.04.14.7, VIT.02.14.1, VIT.03.15.7, VIT.03.16.7) and the Research Council of Norway, grant 240065. J.V. and L.N. were supported by the Czech Science Foundation under Project 20-13616Y. The publication charges for this article have been funded by a grant from the publication fund of UiT The Arctic University of Norway.

*Data availability.* The data to reproduce the rocket measurements in figure 7 can be obtained from the UiT Open reseach Repository at https://doi.org/10.18710/N8GF1U. The relevant data set is tagged 'M1BP_m2'.

*Competing interests.* The authors declare that they have no conflicts of interest.



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
