# Peer review of "A Comparison of Contact Charging and Impact Ionization in Low Velocity Impacts: Implications for Dust Detection in Space"

_Annales Geophysicae, 2020_

## Referee Comment (RC1) · Anonymous Referee #1 · 15 Jun 2020

This paper describes a model for impact charge generation at slow speeds (< 10 km/s) that relies on contact charging through capacitive coupling. The results of this paper are novel and of substantial interest to the hypervelocity impact and cosmic dust detection communities, and introduces physics that were not previously captured by charge generation models based purely on shock induced ionization. The paper is generally well written and easily understood, with comparison to experimental results in literature that corroborate the model's validity over the impact parameters of interest.

However, I would like to suggest several revisions to clarify the derivation of the model and the uncertainties involved in the comparisons presented with respect to the shock

ionization model and experimental results.

First, in Section 2.1, Equations 1 and 2 are stated to be from Wang and John (1988) but the equations do not appear in this form in the cited paper. Certain assumptions appear to be made implicitly, such as the Poisson's ratio needed for the relation between $k_i$ and $E_i$, and the form of alpha appears to be inconsistent between Wang and John (where alpha should have units of length for $a^2 = alpha * R$) and this manuscript (where alpha should be dimensionless for $A = alpha * pi * r_p^2$). The derivation of Equation 2 from the model presented in Wang and John is not obvious and could use additional clarification.

Second, in Section 4.1, a comparison is made between the proposed contact charge model, the Drapatz and Michel model, and experimental results by Mocker et al. It is concerning that in Mocker's paper the Fe on Ag impacts for the charge production power law only included impacts greater than 4.7 km/s, but is extrapolated here in Figure 9 to less than 1 km/s to demonstrate where the Saha-Langmuir solution breaks down. While it is understandable that experimental data is limited, the manuscript should state more explicitly where assumptions are being made about the validity of extrapolated empirical laws, especially when the focus of the paper is on the transition to different physics becoming dominant at low impact speeds. Similarly, more detail on the MUDD measurements and the method used to produce the simulated results shown in Figure 7 would be of value in considering whether this model performs better than prior models such as Drapatz and Michel. Is the simulated current based on the overall dust density values, assuming an average impact rate? Or is it derived from a set of individual impact events?

If these two concerns are addressed, I believe that this paper would be worthy of prompt publication. However, in addition to the two prior concerns, there are a number of typographical errors and similar minor concerns that I enumerate below.

1) Page 2 line 22: At the end of Section 1, the outline should include a brief description

of Section 4.2 and possibly Section 5 for completeness.

2) Page 2 line 28/29: The motivation behind *the* model [...] the *critical* limit

3) Page 2 line 32: Should it say that "models such as shock wave ionization *underestimate* the produced charge"?

4) Page 3 line 26: only a small part of the projectile *contributes* to...

5) Page 4 line 2: Perhaps define MSP in 1st paragraph of Section 2.1 where meteoric smoke is first mentioned.

6) Equation 3: Define units, specify whether the distribution is cumulative

7) Page 4 line 15: More specifically, please specify "... use a packing efficiency value of 70%", and perhaps define the abbreviations used for the lattice structures

8) Page 4 line 18: ...as discussed in *Section 3*.

9) Page 5 line 1: Could the 5-10 factor be attributed to uncertainties/errors in the fill factor or other parameters, rather than only volume fraction contributing to the measured charge?

10) Figure 2: Size distribution of *parameterized* MSP particles

11) Page 6 line 11: The use of $Z_e$ here for a length variable is confusing given the prior use of $Z_{tot}$ for charge. Can a different letter be used for the separation distance?

12) Page 6 line 24: decreasing

13) Page 6 line 25: lowercase yield stress

14) Table 1: lowercase stainless steel, and please use a different abbreviation (maybe SS) to avoid ambiguity with the Saha-Langmuir solution later in the paper. SI prefixes could also be used in the 2nd and 3rd row headers for the table.

15) Page 7 line 18: Specify coulombs for |e|

16) Page 7 line 21: utilize

17) Page 8 line 1: ultimately

18) Page 8 line 12: the most widely cited velocity exponent is ...

19) Page 8 line 21: described

20) Page 8 line 25: we assume that the ice... (delete "for")

21) Page 8 line 29: may be motivated by the typical size...

22) Page 9 line 7 and 11: PAH and BCC have not been defined on first use

23) Page 10 line 16: the fragmentation model results *are*

24) Page 10 line 17: its value are two orders ... (delete "to")

25) Page 10 line 32: Perhaps include some discussion on the origin of pre-charge on particles in space and in an experimental facility

26) Figures 4-6: The use of a mass-specific unit on the left (C/kg) and absolute on the right (Z) is a bit confusing.

27) Page 13 line 21: shaded area shows values of (delete "a")

28) Page 16 line 21: The sentence here can be rephrased to read more clearly. Perhaps reverse the order to read "Assuming the particle has bulk properties, which is suitable [...], the available amount of substance [...] has a volume of [Equation 11]

29) Equation 12 and Page 16 line 28: Perhaps K should be italics (mathtype) to distinguish from K used for potassium

30) Figure 8: "the solid red line *shows*", and perhaps use a consistent colour scheme for traces derived from experimental results, black for contact charge model, etc.

31) Page 18 line 7: as *it* introduces

32) Page 19 line 12: Can the lower speed range of beta meteoroids be quantified here?

33) Page 20 line 15/16/19: Please be consistent with using ESO as the abbreviation after first use

34) Equation A3: Is the final term in the second summation correct here? It looks like the rˆ2 term might have been retained from the previous equality

25) Equation A5: Alpha has a dependence on (vˆ2)ˆ(2/5) - should the 4/15 here be 4/5?

26) A number of the references have a broken link with a duplicated URL header (the "https://doi.org" part)

---

## Referee Comment (RC2) · Anonymous Referee #2 · 13 Jul 2020

This manuscript presents model the combine the fragmentation of microparticles upon impact with a contact charging model, which is in turn compared to more conventional impact ionization model. I have a set of more general comments followed by specific comments. Based on the number and the of depth of the comments, I believe it is best for the manuscript to undergo a major revision and resubmitted.

General comments:

(1) The manuscript is somewhat difficult to follow and it is easy for the reader to get lost in the detail of the large number of assumptions made and models discussed. The recommendation is providing a high-level description of the charging model that is

straightforward to understand even for readers not immediately familiar with the field of impact ionization and/or the challenges of Faraday cup measurements from sounding rockets. This should be followed by the discussion of the details, including the arguments for the validity of the model over the entire range of parameters. An example of such parameter would be the validity of the model over the relatively large velocity range of 0.1 – 10 km/s, which I had a hard time to comprehend.

(2) The manuscript is missing an unambiguous description of the similarities and differences between the model introduced (fragmentation/capacitive charging model) and the more classical impact ionization model that it introduces. This has been confusing to me throughout the manuscript. In impact ionization, free charges are generated in the form of electrons and positively and negatively charged ions. The total charge of electrons and negative ions roughly equals that of the positive ions. The key from above is the that charges are in a free form, meaning that they can be extracted from the impact plasma. In comparison, the fragmentation model seems to keep the charges on the distribution on the fragments, i.e. not in a form of electrons and anions/cations. On the contrary, the contact charging between two materials would provide the particles preferentially with one polarity. It is not clear whether the manuscript is comparing apples to apples, meaning that the fragmentation model is really capable of describe some of the basic properties of impact ionization.

(3) This point is a follow-up for the comment above. As known and demonstrated, impact ionization also provides the means for analyzing the composition of dust particles. This is because the ions extracted from the impact plasma are characteristic to the composition of the dust material (and also that of the target). This fact has been greatly ignored by the manuscript, while one could argue that there is tremendous information in the composition of such ion mass spectra as a function of velocity. This has been investigated to detail by several authors, most notably by Fiege et al. and Hillier et al. Any model that is to update impact ionization needs to be consistent with such findings. It is recommended that the manuscript is extended by such discussion

(4) It appears that the manuscript is trying to apply the model over too large of a parameter space. The manuscript claims that the model applies over a wide impact speed range (0.1 – 10 km/s) and for a range of particles materials; from including icy grains metallic particles. I have a hard time believing that this is realistic (e.g. the normalized impact energy spans over 4 orders of magnitude). For example, at the low end of the speed range, a metallic particle could simply just deform (as a bullet from a rifle does). Is it possible that the model has enough free parameters that it can fit basically any experimental data, regardless of its physical validity? The recommendation here is provide a more detailed and more focused discussion why such model should be valid over such a wide range, and/or specify more carefully to what scenarios the fragmentation model is applicable to. (Apologies if I greatly misunderstood the manuscript. In this case, see comment #1.)

Specific comments:

- The abstract reads like a description of what the paper is about. Generally, the abstract is expected to be a short version of the paper. Revision is recommended. - Top of page 2. On the applicability of the Saha equation: To my best knowledge, every study on this topic has found Saha equation to be not applicable for impact ionization. Other than of course that (1) Ions of low ionization species have a large relative abundance in the impact plasma, and (2) elements (molecules) appear to have a velocity under which they are not found in the analyzed impact plasma. See, for example, the Fiege et al. papers. A revision is recommended. - Top of page 2: I believe the authors meant Auer 2001, instead of 2012. - Page 2: It would be good to clarify what contact charging means, meaning that fragments become charged? How does this comply with the current knowledge of impact ionization in terms of generating free charge carriers (electrons, atomic and molecular ions) and their composition, as discussed in the general comments? - Contact charging (or triboelectrification) has been extensively studied in the literature and providing some of the relevant reviews would be useful for the reader. - Section 2.1. The claim of the fragmentation of small particles (> 10 nm) is based on

two cited work. Froeschke et al. discusses the fragmentation of nanoparticle agglomerates. Tomsic et al. described the fragmentation of molecular ice clusters. Since both these works describe the fragmentation of rather fragile systems, it is not clear whether is truly appropriate to extrapolate these results to the fragmentation compact and refractory dust particles (e.g. metals, or rocky/mineral fragments), and over such large velocity range. The recommendation is to strengthen the arguments by providing a more detailed discussion. - Section 2.1. Sub-nano-scale particles are mentioned. The electronic properties consisting of tens or hundreds of atoms/molecules may be different from those of the bulk properties. For example, the work function. Is there size limit of validity that is worth mentioning? - Page 3. The paper claims to apply Hertz's law for elastic deformation is assumed 'for the impact energies encountered in this paper'. Not clear whether this is meant to apply for the entire 0.1-10 km/s impact speed range as stated in other parts of the paper. Please clarify. - Fig. 1 and relevant text: It was not too clear what is happening in this model. Is the particle only deforming or also fragmenting? Maybe I have missed it, but in any case, a high-level model description would be useful for the readers. - Page 4: Bin width of 0.01 nm (i.e. 10e-11m). That is smaller than the size of an atom and appears non-physical? Please clarify. - End of p. 4: $Z_{tot}$ charge – what sort of charge is this? Free or bound to the fragments? Is this the charge exchanged between the particle and the surface summed up for all fragments? Please clarify. - Wouldn't it be reasonable to assume that the fragmentation of the particle has a strong dependence on impact velocity? - Section 2.2. The expression of capacitance appears to have incorrect unit. This likely should be $C = eps\ A/Z$, rather than $eps\ A\ Z$. - Section 2.3. The two velocity regimes (5-10 km/s and > 10 km/s) are quoted as the two regimes of shock wave ionization. This is somewhat confusing wrt the work by Mocker et al., where the velocity threshold has been shown for which volume ionization is epexted to dominate. Which is, btw, outside of the interest of this manuscript that is limited to < 10 km/s. Can the authors provide some level of discussion and reference where for the two velocity ranges for shock-wave ionization are discussed? - Figure 4. Is it reasonable to assume that the work function of fragments

as small as 0.2 nm will be similar to that of the bulk material? - Figure 4. This figure is confusing. How can the model assume that the particles are the same size (30 nm) throughout the entire velocity range shown? The nature of the accelerator's operation is such that larger particles have lower velocities and smaller particles can reach higher velocities. The normalized yield in the units of C/kg that considers the assumption that the charge yield scales with the mass. It is not clear if the fragmentation model would apply for the varying dust size of the experimental results. Has this been investigated? Is this figure really comparing apples to apples? Please clarify and provide the relevant discussion.

Tidbits:

- Is the quantization of the charge on the fragments considered? Small particles can carry only one elementary charge (or zero) and the total charge would be the statistical average over all particles. Please clarify or discuss as appropriate. This comment is relevant to equation A4 as well.

- It might be useful to note that the literature has already discussed the impact ionization of icy dust grains in relevance to the Cosmic Dust Analyzer measurements in Enceladus's plumes. Please see the papers by Postberg and Abel (some combination of these two authors, plus other coauthors). Basically, it has been found that the charge production in this case is best described by a MALDI-type process, where the preformed ions in the icy matrix are released by the impact of the particle and the evaporation of the ice. It might be beneficial to check the validity of the fragmentation model against these findings.

---

## Author Comment (AC1) · 14 Sep 2020

"A Comparison of Contact Charging and Impact Ionization in Low Velocity Impacts: Implications for Dust Detection in Space" as submitted by Antonsen et al. to ANGEO

We thank the referee for their time to review the manuscript, and for a number of helpful and important comments. In the following, we have tried to respond to all of them, and we have presented the revisions we have carried out in connection to the respective comments. We have not made any major revisions in the overall structure of the paper as both referees seem to agree with the structural setup. Major revisions in the content have been made, and we also refer to the open response to Referee 2. We note,

especially, that we have given a better overall description of the mechanisms discussed in the paper. We have also revised the abstract.

Regarding our equations (1) and (2): We have used the wrong citation here. The correct source is John et al (1980), and this has now been corrected. We have also revised the citations in the first paragraph of section 2.2; Here we have now stated that we use the general theory of John et al. (1980) and the results of Wang and John for plastic collisions. The alpha-parameter is, as very correctly stated, not trivial to derive and we have relied on Soo's (1970) utilization of Hertzian deformation. Have added a reference to this work below the eqs.

Regarding the referees second comment (range of velocities in Mocker et al experiments and extrapolation of results): This is a very good point (raised by both referees), and is due to a poor description of our utilization; The reason why we have used Mocker et al. results is that the quality of the data is very good in their range of interest. What is not clearly stated here, is that their result is almost "indestinguishable" (for our purpose; Fe-on-Ag) from the findings of Collette et al 2014, who have investigated impacts done to speeds of 2 km/s – which is well embedded into the range of velocities we need comparison data for in order to compare contact charging and impact ionization. Therefore we have used Mocker's result, however, we could have used Collette et al. with identical conclusions. The data from both Mocker et al and Collette et al for speeds below 10 km/s is furthermore not in the volume ionization regime, which is only expected when impact energies exceed the Fermi-limit (several tens of km/s). Revision: Have tried to specify why we have chosen the Mocker et al results in the start of paragraph 2 of section 3.2.

Regarding the MUDD-results and Figure 7: A very helpful comment. The simulated current is based on high-resolution dust density and size (average size, monodisperse) data as obtained by Havnes et al (2019). This is referenced in the last paragraph of the section. Due to the very low impact velocity, for reasons discussed in section 4.2., we felt the inclusion of the Saha-Langmuir solution in figure 7 was unnecessary. In fact,

even the pre-charges on the ice particles alone would completely dominate the current in MUDD in comparison to shock wave ionization, in the case of rocket. We agree with the referee that the text discussing figure 7 can be improved, and have therefore implemented a revision of this. Hopefully, the text reads better.

Minor comments: For corrections of typos and insertion or removing of certain words, we have not listed the revision below. (1) Revised as suggested. (3) Yes, underestimate. (5) Defined MSP. (6) Revised. (7) Added explanation and abbreviations. (9) Yes, there might be an additional factor there, but lowering the filling factor into the lower end around 1%, there is still a significant overestimation present. (11) We have changed Z to capital Delta. (25) Added descriptions of pre-charge. (26) This was chosen due to the "common" practice, as well as it allows for direct comparison between the results. As far as the contact charging model goes, since it is specific yield (C/kg), it does not matter whether or not the incoming (modelled) projectile is 30 or say 100 nm, since – as shown in the appendix – $Q_p$ is proportional to $r^3$. Thus, the specific yield is the same for any size. A difficult question is then how the plasticity and other modelling parameters change with velocity, which can become complex to give a thought out answer to. (28) Rephrased. (32) The speed range of beta meteoroids is still not very well known due to the fact that we are not certain about where the formation region lies. Even at present, different authors use formation radii in the range 5-20 Solar radii, which would yield very different velocity ranges (assuming only conservation of energy and angular momentum control the orbits of the meteoroids). We hope Parker solar probe and solar orbiter will answer open questions about beta meteoroids and their velocities.

(35) It is raised to the power of 1/3 after, so 4/5 -> 4/15.

Please also note the supplement to this comment:
https://angeo.copernicus.org/preprints/angeo-2020-23/angeo-2020-23-AC1-supplement.pdf

---

## Author Comment (AC2) · 14 Sep 2020

Response to Referee 2, handle angeo-2020-23-RC2 "A Comparison of Contact Charging and Impact Ionization in Low Velocity Impacts: Implications for Dust Detection in Space" as submitted by Antonsen et al. to ANGEO

- Please see the supplement PDF for proper typeset and sectioned response.

We thank the referee for a number of helpful and interesting comments. In the following, we have tried to respond to all of them, and we have presented the revisions we have carried out in connection to the respective comments. We have not made any major

revisions in the overall structure of the paper, only major revisions in the content. We have also revised the abstract.

General comment 1: We understand that the current presentation of the theory is rather complicated, and may benefit from some revision. Firstly, we strongly agree with the referee that the validity of the model should be discussed in a better manner. We believe that this would also clear up some of the confusion the reader might have. We address this issue first, below, and address the point of a high-level description as brought up by the referee after that. 1. The referee brings up an important point here – the discussion of the validity of the model over the range of parameters used. The intention of utilizing a contact charging model is that it is only really representable for the lower part of the velocity spectrum introduced in the paper (0.1 to 10 km/s). This is not the intention either. What we have failed to convey, based on the confusion brought up by the referee, is however that there is a change in charging mechanisms between traditional impact ionization (Langmuir-saha) and capacitive charging. To say however, exactly where the overlap between these two regimes finds place is difficult both experimentally and theoretically. In fact, we do not think there exists an answer in the literature currently that explains the sudden change ("discontinuity") between charge mechanisms at O(1 km/s) impact speeds theoretically, and it is beyond the scope of this work to answer that question. Revision: It was intended that the discussion in section 4.1 shall address the validity of an impact ionization vs. contact charging model at 'low' speeds, but we can agree that this discussion is presented too late in the work. We have therefore revised the manuscript by adding a paragraph already in the introduction noting that the current work does not address the exact limit (or speed) that a change between mechanisms happen, only that it happens within the range and that we discuss the arguments for validity of one mechanism before the other [P. 2, L. 19]. Furthermore, we feel that the introduction of section 2 introduces the details of the contact charging—impact ionization dichotomy well.

2. We feel that the essence of Section 2 provides a good high-level introduction into
the problem at hand and our solution to it, however, we agree with the referee that the current presentation of the material is not optimal. We feel that a better high-level description/introduction can be given by improving Section 2. Revision: At the very beginning of Section 2, we have added what can be considered a high-level 'simple' explanation, which we hope serves the purpose of guiding non-experts towards an understanding of the deeper topics of the paper.

General comment 2: The referee raises another well founded and important point, which implies that the presentation or rather distinction between the two addressed modes of charging is not sufficient. To clarify, what was intended understood from the manuscript: The paper is indeed not comparing apples to apples, as the two discussed mechanisms are fundamentally different. Also addressed in 'comment 1', the two mechanisms in fact compete. To describe the difference at a deeper level than currently, we believe will make the work considerably more complex to comprehend (and probably more speculative). There is no presence of ionization or dissociation at a fundamental level in our contact charging mechanism. Thus, the referee states correctly that the charge production is closely bound to the distribution of fragments. It is also correct that one polarity is completely dominating – which is exactly what has been consistently shown at many instances for rocket experiments and low velocity ice-on-metal collisions in the lab (e.g. referenced works by Havnes, Næsheim, Antonsen et al., Tomsic et al. and so forth). We feel it is also important to point out that the mass spectral characteristica which can be drawn out from conventional impact experiments should not be discussed in connection with our charging model. However, it is merely a tool to better understand when shock wave ionization of certain materials becomes effective/dominant – as we have utilized it in the manuscript. Revision: In addition to the changes made in connection to general comment 1, which we believe clears some confusion; we have added at the end of the first section of the first paragraph of section 2 [p. 3] an elaboration of the fundamental differences between the discussed mechanisms. This paragraph is subsequently followed by a motivation that we now feel is easier to comprehend and should explain the essence of our modelling efforts better.

General comment 3: As is hopefully much more clear from the responses above and corresponding revisions in the manuscript, our contact charging model not intrinsically equivalent to ionization as it happens in impacts of small particles with solids. It therefore does not make sense to discuss mass spectral properties of the impact debris of the fragmentation cloud which we provide the model for. There are two important points in the paper where we do in fact touch upon issues related to the role of ions in the impact cloud: 1. In the discussion of the overlap (in produced charge) between impact ionization and contact charging, where it makes sense to note where the literature finds onset of specific ions in impact gas mas spectra. This is interesting, because it is a rough indication of when (shock) impact ionization becomes dominant, and we consequently mention this in section 4.0. We have here relied on work done with the dust accelerator in Heidelberg, and we find the quality and amount of that data to be sufficient for our purpose. 2. In the comparison of our proposed mechanism to the low velocity shock wave ionization solution as presented by Drapatz and Michel. In that regime (i.e. below the limit where mass spectra can detect significant ion partial pressures), particularly volatile ions (e.g. of alkali metals) can diffuse through the molten fragments (or droplets, as Drapatz et al. refers to them) and can be released from their surface. The degree of ionization and produced charge can subsequently be described well with a Saha-Langmuir equation. We address this in depth in section 4.1. We also want to stress that we do value the tremendous effort behind and results from the Cassini CDA. As the TOF mass spectral properties are not very useful for the direct comparison with our contact charging mechanism, as we have tried to convey in the manuscript, we have not focussed on these. However, from the good advice given by the referee, we understand that a mentioning of such works can easily be justified and have included a reference and description of Hillier et al CDA results in section 4.

General comment 4: The main issue in this point was addressed in the response to general comment 1, above. We hope it is much clearer as to which regimes of validity we are interested in. As a side note, the mid-range velocities in which shock wave ionization have been proposed to dominate in (some km/s to a few tens of km/s)

have the same velocity-to-charge relationship, or rather power-law, through the whole range, i.e. Q~mxvy. Thus, a somewhat arbitrary upper velocity limit of 10 km/s was chosen. To exaggerate the argument: the limit could have been set to the arguably more arbitrary 9 km/s or 12 km/s – it does not really matter for the presentation of our results. From the discussion in section 4.1, we hope it is clear that we do not propose that contact charging is effective in the upper part of the investigated velocity range. The referee makes a very good point in that the Hertzian deformation theory which we assume cannot be valid at the upper higher speeds investigated here. We have tried to convey this message throughout the paper, but since it may be unclear, we have also emphasised this issue in the presentation of the fragmentation model in section 2.1. The referee mentions the extreme that at the low end that a grain could behave like a bullet – simply deforming. This is not possible, as the elastic-plastic crossover for grains smaller than several tens of nanometres happens as much lower speeds (even <100 m/s) than interesting for this work [e.g. Rennecke and Weber 2014, Froeschke et al., Tomsic et al.]

Specific comments: Smaller adjustments; re-phrasing, typos, reusing citations is not addressed below, however, we have tried to follow the recommendations of the referee.

Regarding Saha-solution applied in impact ionization: The referee is correct that a pure Saha-solution is not necessarily valid. What we (and originally Drapatz and Michel) utilize for the low velocity limit of shock wave ionization is the modified Saha-Langmuir equation (now revised in the abstract). This is a Saha equation that bound electrons in metals are indeed distributed. Such a solution has been shown to hold for impurity ionization in impacts below ~10 km/s, and we present the basics in section 2.3.

Regarding Auer 2012: This is a newer/revised edition (?) It is the preferred citation as listed from Springer Online, at least.

Regarding contact charging: We feel this is now sufficiently answered through the revisions in connection to the general comments above.

Regarding section 2.1: A good point. The citations, as we feel can be understood from the text is used to back up the claims about the degree of fragmentation. Other references in the section help create a more rigorous literature background. An important point is also that key parts of the theoretical background/literature cited in connection to the fragmentation is discussed in section 3.1 (in the Results) where we feel it is much more natural to bring up. Revision: This is referenced to in section 2.1, however we have adjusted the first paragraph of section 2.1 to help the reader understand the grounds on which we have chosen modelling parameters.

Regarding Hertzian deformation: revised.

Regarding figure 1: Revised overall description in the text and in the figure caption.

Regarding binning: Misprint. Binning is done with 1 Å spacing, but the sentence was removed as it does is not matter for the end result; a continuum binning ( say 0.000. . .01 nm) would yield very similar results when the fragmenting particles is several nanometres.

Regarding section 2.3: a very good point, brought up by both referees. The reason why we have utilized Mocker et al. results is that the quality of the data is very good in their range of interest. What is not clearly stated here, is that their result is almost "indestinguishable" (for our purpose; Fe-on-Ag) from the findings of Collette et al 2014, who have investigated impacts done to speeds of 2 km/s – which is well embedded into the range of velocities we need comparison data for in order to compare contact charging and impact ionization. Therefore we have used Mocker's result, however, we could have used Collette et al. with identical conclusions. The data from both Mocker et al and Collette et al for speeds below 10 km/s is furthermore not in the volume ionization regime, which is only expected when impact energies exceed the Fermi-limit (several tens of km/s). Revision: Have tried to specify why we have chosen the Mocker et al results in the start of paragraph 2 of section 3.2.

Regarding Ztot: Bound on fragments. Revised.

Regarding work functions: This is mentioned in the end of the discussion as one of the more uncertain parameters/assumptions, as the work function for nanoscale particles is in fact size dependent to some extent (ref e.g. Wood 81). From mesospheric studies, e.g. Plane et al., it is not necessarily such a large difference for aerosols at sizes of nanometres to tens of nanometres, that it makes a very large difference. Tens of nanometres is by rule of thumb "bulk" as far as cohesive and surface binding energies goes and thus also work function. However, as has also been mentioned, if all modelling parameters are tuned to the extremes of their tolerances, the resulting charge production could change significantly.

Regarding last comment on figure 4: Another very good point that we have had to think about. As far as the contact charging model goes, since it is specific yield (C/kg), it does not matter whether or not the incoming (modelled) projectile is 30 or say 100 nm, since – as shown in the appendix – $Q_p$ is proportional to $r^3$. Thus, the specific yield is the same for any size. A difficult question is then how the plasticity and other modelling parameters change with velocity, which can become complex to give a thought out answer to. We feel it lies a bit outside of the scope of the paper, as the current work is "only" meant to show that we need a better description for the "low velocity" range of impact charging and that our model is a good way to explain some of the observed phenomena connected to such impacts.

Tidbits: - Charge is not quantized at this point, which may or may not overpredict the contribution from the smaller fragments. Effectively, the charging current can be viewed as a charging probability. A resolution of this 'issue' must also discuss in details the role of potential wells on the surface of the smallest fragments – which is probably a phenomenal modelling effort.

- MALDI-type processes are certainly interesting, although may lie a little bit outside our parameter range of interest. In general, the speeds necessary to obtain complete evaporation and have energy available to facilitate MALDI are not obtained at the impact energies we discuss here.
Please also note the supplement to this comment:
https://angeo.copernicus.org/preprints/angeo-2020-23/angeo-2020-23-AC2-
supplement.pdf